# Immune Dysregulation and Trophoblastic Dysfunction as a Potential Cause of Idiopathic Recurrent Pregnancy Loss

**DOI:** 10.3390/biology14070811

**Published:** 2025-07-04

**Authors:** Sara Vasconcelos, Ana Costa Braga, Ioannis Moustakas, Bruno Cavadas, Mariana Santos, Carla Caniçais, Carla Ramalho, Susana M. Chuva de Sousa Lopes, Cristina Joana Marques, Sofia Dória

**Affiliations:** 1Genetics Unit, Department of Pathology, Faculty of Medicine, University of Porto, 4200-319 Porto, Portugal; svasconcelos@med.up.pt (S.V.); marianasofsantos@outlook.pt (M.S.); carlacanicais@gmail.com (C.C.); cmarques@med.up.pt (C.J.M.); 2RISE-Health, Department of Pathology, Faculty of Medicine, University of Porto, Alameda Prof. Hernâni Monteiro, 4200-319 Porto, Portugal; 3Department of Pathology, Centro Hospitalar Universitário São João (CHUSJ), ULS São João, 4200-319 Porto, Portugal; 4Department of Pathology, Faculty of Medicine, University of Porto, 4200-319 Porto, Portugal; 5Department of Anatomy and Embryology, Leiden University Medical Center, 2333 ZC Leiden, The Netherlands; 6Sequencing Analysis Support Core, Leiden University Medical Center, 2333 ZC Leiden, The Netherlands; 7i3S—Instituto de Investigação e Inovação em Saúde, University of Porto, 4200-135 Porto, Portugal; 8Department of Obstetrics and Gynaecology, Faculty of Medicine, Centro Hospitalar Universitário São João (CHUSJ), ULS São João, 4200-319 Porto, Portugal

**Keywords:** pregnancy loss, transcriptome, cytotrophoblast, syncytiotrophoblast, extravillous trophoblast, immune dysregulation

## Abstract

Recurrent pregnancy loss without a known cause can be deeply distressing for couples and remains a challenge in clinical care. In this study, we analyzed tissue from early pregnancy losses to uncover biological reasons that might explain why these miscarriages happen. We found that certain important cells in the placenta, responsible for supporting the pregnancy and producing hormones, were reduced. At the same time, cells related to immune function were increased. We also observed signs of inflammation, suggesting that the mother’s immune system might play a role in these losses. These results point to problems in how the placenta develops and how the immune system responds during the early pregnancy. Our findings offer new insights into why some pregnancies fail without explanation and may help develop better diagnostic tools or treatments in the future.

## 1. Introduction

An estimated 50% of all conceptions are lost at preclinical stages, including biochemical loss and implantation failure, with an additional 9–20% of clinically recognized pregnancies resulting in miscarriage, primarily during the first trimester (weeks 5–12 of gestation) [1,2]. Gestational loss occurring up to 12 weeks of gestation is classified as early gestational loss [3] and may occur as either a sporadic or a recurrent event. Recurrent pregnancy loss (RPL) affects 1–5% of fertile couples and represents one of the most complex and challenging issues in reproductive medicine, often associated with significant psychological distress [4]. This condition is defined as failure to progress of two or more clinically recognized pregnancies, according to the guidelines from the European Society for Human Reproduction and Embryology (ESHRE) [5] and the American Society for Reproductive Medicine (ASRM) [6].

Several risk factors for RPL have been identified, including advanced maternal age, obesity, congenital or acquired anatomical defects of the uterus, abnormal parental karyotypes, thrombophilia, infections, autoimmune disorders, as well as endocrine and endometrial dysfunction [1]. Notably, fetal chromosomal abnormalities represent the most frequent cause of pregnancy loss, particularly of early pregnancy loss, accounting for up to 50% of cases. However, the incidence of fetal chromosomal abnormalities in RPL decreases with an increasing number of pregnancy losses [6,7]. Approximately half of pregnancy losses present a normal karyotype, with a genetic cause remaining unidentified. Currently, few interventions and treatment are available for RPL, and there remains a significant challenge in elucidating the molecular mechanisms underlying RPL [5,6,8].

The first trimester of pregnancy is a critical period for ensuring a healthy pregnancy, requiring effective placentation for a successful outcome. The placenta plays a crucial role in sustaining the growth and development of the embryo, participating in essential processes such as early embryo implantation, vascular remodeling, and immunological tolerance. The formation and development of the placenta is a rigorously regulated process that is closely linked to the proliferation, differentiation, and functional state of placental trophoblasts [9]. The human placenta is characterized by the extensive invasion of trophoblasts into the maternal endometrium at the maternal–fetal interface, which lies between the uterine mucosa and the extraembryonic tissues of the developing conceptus. This invasion enables direct contact of the trophoblasts with maternal blood [10]. Trophoblast invasiveness is regulated by a complex network of cell types, mediators, and signaling pathways. During this critical process, the trophoblast lineage develops at the time of blastocyst formation and divides into two main cell populations: the villous trophoblast, comprising villous cytotrophoblasts and the multinucleated syncytiotrophoblast (which forms the outer layer of all placental villi), and the extravillous trophoblasts, which invades maternal uterine tissues [11,12].

In the earliest stages of pregnancy, multinucleate syncytialized cells, referred as primitive trophoblast, are observed at the invasive leading edge of the implanting embryo. The proliferation and invasion of extravillous trophoblast cells into the decidua and uterine myometrium are essential for remodeling of uterine spiral arteries and for establishing maternal–fetal circulation [13]. To achieve maternal–fetal tolerance and successful placentation, an intricate interaction is established between fetus-derived trophoblasts, maternal immune cells, and decidual stromal cells during a normal pregnancy. Decidual immune cells not only maintain immune tolerance but also regulate trophoblast function to support fetal development [14,15,16]. Considering all these important placental roles, it is believed that RPL may be mediated by placental trophoblastic dysfunction or by deficient formation of the maternal–fetal interface [17,18,19]. The pathways affected in pregnancy loss may be linked to this interaction, either directly or indirectly, during a time window when endometrial and chorionic tissues are rapidly changing their architectures and microenvironment [20]. The highly variable clinical manifestations and the complexity of RPL etiopathogenesis makes it challenging to predict and prevent its occurrence. Hence, there is an urgent need to identify the molecular mechanisms underlying trophoblast turnover in RPL and identify corresponding biological targets, towards novel therapeutic strategies. Although multiple factors have been implicated in pregnancy loss, the precise cellular and molecular mechanisms underlying idiopathic pregnancy loss (iPL) remain largely unknown. Most existing studies describe transcriptomic alterations in iPL samples but do not explore what these changes imply in terms of cellular identity, trophoblast differentiation, or functional roles.

## 2. Materials and Methods

### 2.1. Ethical Approval and Sample Collection

This study was approved by the Health Ethical Committee of the Hospital University Centre São João (CHUSJ)/Faculty of Medicine of the University of Porto (FMUP) (project number 430/19). We collected nine human products of conception (POC) from idiopathic pregnancy loss (iPL), in the first trimester of pregnancy (from 6 to 10 weeks of gestation), in collaboration with the Gynecology and Obstetrics Department of the CHUSJ. The majority of these samples (8 out of 9) were obtained from patients who had experienced two or more pregnancy losses and were hence classified as recurrent pregnancy loss (RPL) (Table 1). The exclusion criteria included the identification of chromosomal abnormalities, pregnancies resulting from assisted reproductive technologies (ART), twin pregnancies, acquired and hereditary thrombophilia, and the diagnosis of infectious, endocrine, or autoimmune diseases. The tissues were dissected under a dissecting microscope, in sterile conditions and washed with sterile PBS to remove excess blood. The villi were preferentially isolated whenever they were visually observed. However, when they were not observed, a representative portion of the entire sample was nonetheless stored for analysis. Samples were then immersed in RNAlater (Invitrogen, Thermo Fisher Scientific Inc., Waltham, MA, USA) and immediately stored at −80 °C for subsequent analysis.

Due to clinical protocols and national regulations in Portugal, it is not feasible to obtain samples from elective terminations of pregnancy (ETP) for this study. Women who choose to terminate a pregnancy for personal reasons are not hospitalized for the expulsion of the product of conception; instead, the procedure is carried out in an outpatient setting, typically at home. As a result, these samples are not collected or preserved under standardized conditions suitable for transcriptomic analysis. Additionally, recruiting individuals from this emotionally sensitive population would entail further ethical and logistical challenges. To address this limitation, we included control samples from more than one previously published dataset, allowing us to include multiple independent control groups.

### 2.2. RNA Extraction from Idiopathic Pregnancy Loss

The tissues were homogenized using Triple-Pure™ zirconium beads (Bertin Technologies, Montigny-le-Bretonneux, France) in 1 mL of TRIzol reagent (Thermo Fisher Scientific Inc., Waltham, MA, USA) with a MiniLysis homogeniser (Bertin Technologies, Montigny-le-Bretonneux, France). RNA extraction followed the manufacturers’ protocol. RNA quantification and integrity were measured with the Agilent 2100 Bioanalyzer (Agilent Technologies, Waldbronn, Germany), with RNA integrity numbers (RINs) ranging from 5.2 to 9.6.

### 2.3. RNA-Sequencing and Gene Ontology Analysis of Idiopathic Pregnancy Loss Samples

Bulk transcriptome analysis was performed as described in our previous study [21]. The normalized counts are shown in Appendix A (Appendix A).

For downstream analysis, the gene expression count table was used as input for an R (v4.1.0) script. Principal Component Analysis (PCA) was performed, and samples were visualized on a scatter plot using the first two principal components. Gene expression was visualized using heatmap plots. Differential Gene Expression analysis was performed using DESeq2, v1.34.0, resulting in a table of differentially expressed genes (DEGs) (Appendix A). Statistically significant DEGs were identified using a false discovery rate (FDR) < 0.05 and were visualized on a volcano plot generated based on the −Log10(FDR) on the y-axis and avgLog2FoldChange on the x-axis. Gene Ontology (GO) term enrichment was performed using the gprofiler2 (v0.2.1), exploring gene functions of DEGs based on three annotated ontologies: Biological Process (BP), Reactome pathway (REAC), and Human Protein Atlas (HPA) analysis (Appendix A). GO terms with an adjusted *p* value < 0.05 were considered significant.

### 2.4. Comparison with Published RNA Sequencing Dataset

We compared the raw sequencing data of our iPL samples with five published control groups from ETP [22,23,24,25,26]. The accession numbers from the five ETP publicly available datasets are as follows: E-MTAB-9203 (8 villous trophoblast samples of 7–8 weeks of gestation), GSE121950 (3 villous trophoblast samples from 8 weeks of gestation), GSE163651 (3 villous trophoblast samples from 7 weeks of gestation), GSE161969 (3 decidua samples from 41 and 60 days) and GSE113790 (3 decidua samples from 7 weeks of gestation). The published data were either in raw format (FASTQ) or aligned using with a different genome build, so we realigned them to GRCh38 human reference using the GENCODE gene annotation (v38), performing the same bioinformatic steps taken with our data to exclude variability.

### 2.5. Single Cell RNA Sequencing (scRNAseq) Annotation and Deconvolution

For cellular deconvolution of bulk RNAseq data, we first performed the re-annotation of a published single cell dataset. For this, using classifications from Pique-Regi et al. (2019) and Suryawanshi et al. (2018) [27,28], clusters were identified by analysis of expressed marker genes; cluster markers are provided in Appendix A (Appendix A). All scRNAseq analysis were carried out in R (v4.3.0) using the Seurat R package (v4.3.0). The published dataset contained the raw counts per cell for each sample, so no alignment was performed. Each sample was analyzed separately, prior to the integration of the data. Analysis of the distribution of the number of detected genes and UMIs (Unique Molecular Identifiers) per cell allowed to define the thresholds to remove low quality cells. Only cells with a percentage of transcripts from mitochondrial genes below 10% and UMIs ranging from 500 up to 6000 UMI were further kept. Additionally, we removed cells detected as doublets using “scDblFinder”.

Integration of both samples was performed with Harmony, using the normalized SCT values from each individual dataset. For the integration, the 3000 most variable features were calculated using the “SelectIntegrationFeature” function. Harmony integration was performed using a PCA reduction with up to 50 dimensions and using the dataset origin of each cell as the distinctive group. The number of neighbors was calculated using the Harmony reduction for up to 20 dimensions. Clusters were identified using the Leiden algorithm, using the “igraph” method for a 0.5 cluster resolution.

Bulk deconvolution was carried out using MuSiC, using the raw counts from the bulk RNAseq data. Only genes present in both scRNAseq and RNAseq were used for the analysis. Deconvolution was performed using the annotation from the scRNAseq normalized by the library size.

### 2.6. Pathway Enrichment and Differential Expression Analysis Between Idiopathic Pregnancy Loss and Elective Termination of Pregnancy

RNA-seq raw count data was evaluated to identify DEGs between iPLs and control ETP samples. Differential expression analysis was carried out in R (v4.1.2) using DESeq2 (v1.34). Genes with FDR < 0.05 and Log2 fold change = |2| were selected to identify significant DEGs across the five independent comparisons (Appendix A). Overlaps between up- and downregulated DEGs were visualized with Venn Diagrams, created using the VIB/UGent Bioinformatics and Evolutionary Genomics tool. Gene-set enrichment analysis was performed on a pre-ranked list of genes (signed log2FC × −log10(padj)) and was carried out in fgsea using molecular ontology database, which allowed identification of Hallmarks—gene sets representing well-defined biological states or processes. An FDR value threshold of <0.05 was applied to identify significant pathways (Appendix A). The GO enrichment analysis of the overlapping DEGs was performed using gprofiler2, with significant GO terms (molecular function (MF), BP and REAC) with an adjusted *p*-value < 0.05 (Appendix A).

### 2.7. Histological Examination and Immunohistochemistry

For each iPL case (with the exception of one unavailable sample), multiple slides were analyzed from Formalin-Fixed, Paraffin-Embedded (FFPE) tissue blocks of various regions, spanning the embryo, yolk sac, gestational sac/membranes, chorionic villi, intervillous space, endometrium/decidua, and the site of implantation. A specialist in fetal and placental pathology analyzed all slides. Slides were obtained from sequential ~4 µm thick slices of each of the FFPE blocks, at room temperature (RT), according to standard histopathological techniques, and stained by hematoxylin–eosin (H&E). A neutral gum was used for mounting. For immunohistochemical analysis, sections from FFPE tissue blocks were deparaffinized and hydrated, using standard procedures. Primary antibody for CD68 (mouse monoclonal, Kp1, Cell Marque, Rocklin, CA, USA) was used in combination with the OptiView DAB IHQ detection kit according to the manufacturer’s instructions on a VENTANA benchMark ULTRA platform (Ventana Medical Systems, Tucson, AZ, USA). Adequate positive control was included. A brightfield microscopy examination of the slides were performed.

## 3. Results

### 3.1. Transcriptome Profiling of Idiopathic Pregnancy Losses Revealed Two Distinct Clusters

To evaluate how the nine products of conception (POC) from idiopathic pregnancy losses (iPL) clustered together we constructed a heatmap and a PCA plot (Figure 1A,B). The first principal component (71.33% of variance) of the Principal Component Analysis (PCA) revealed two distinct clusters, designated as Cluster 1 (C1) and Cluster 2 (C2) (Figure 1A,B). In total, 7612 differentially expressed genes (DEGs) were observed between the two clusters (FDR < 0.05) (Appendix A). Of these, 3454 DEGs were downregulated, and 4158 DEGs were upregulated in the C2 (Figure 1C). Gene Ontology (GO) and Reactome (REAC)-related pathway analysis indicated that downregulated DEGs in C2 were primarily involved in cell cycle and metabolic processes (Figure 1D). On the other hand, the upregulated DEGs in C2 were associated with the regulation of biological processes, system development, signaling and cell communication, and regulation of response to stimulus. REAC pathway analysis further revealed that the upregulated DEGs were involved in ‘Platelet activation, signaling and aggregation’, ‘Immune system’, and ‘Extracellular matrix organization’ pathways (Figure 1E). Moreover, data from the Human Protein Atlas (HPA) indicated that the upregulated DEGs in C2 were enriched for ‘endometrium’, whereas the downregulated DEGs were enriched in ‘placenta’. These findings led us to hypothesize that our collection of POC samples comprises both villi (enriched in C1) and decidua (enriched in C2). To further test this hypothesis, one female sample of each cluster was compared to matched maternal blood, using quantitative fluorescent PCR of short tandem repeats (STR) markers. The C1 female sample displayed a distinct genetic profile compared to the maternal sample, whereas the C2 sample shared the maternal profile. These observations supported our hypothesis that C1 consists primarily of villi, while C2 is enriched in maternal decidua.

### 3.2. Transcriptomic Analysis of Idiopathic Pregnancy Loss Versus Elective Termination of Pregnancy

Subsequently, we compared the bulk sequencing data from iPL samples with publicly available datasets from elective termination of pregnancy (ETP) controls. Specifically, C1 samples (n = 4) were compared with three villi (VT, Villi Trophoblasts) control groups (n = 8 VT ETP 1; n = 3 VT ETP 2; n = 3 VT ETP 3), and C2 samples were compared with two decidua (DB, Decidua Basalis) controls groups (n = 3 DB ETP 1; n = DB ETP 2) (Figure 2).

The PCA of all samples separated the villi from decidua samples along PC1 (44.7% variance), and samples clustered according to their respective studies, except for one sample in DB ETP 1 group, which was identified as an outlier (Figure 2A). In the villi control groups, the VT ETP 3 samples were more distant from the others, possibly due to methodological differences, namely the use of RPMI culture media with supplements for washing and culturing villi samples, as well as the inclusion of an initial cultivation step on collagen-coated plates prior to isolating extravillous rather than villous cells. All control groups (except VT ETP 3) clustered towards the top of the PCA plot, while the iPL samples clustered at the bottom.

To assess the cell composition, we performed cellular deconvolution of the bulk RNAseq data. The single cell transcriptome re-annotation analysis of a published dataset, guided by established markers, revealed distinct cell clusters. The TSNE annotation of the clusters was performed, and eight distinct cell types were noted, namely, cytotrophoblasts (CTBs), syncytiotrophoblasts (STBs), extravillous trophoblasts (EVTs), erythroblasts (EBs), natural killer (NK), Hoffbauer cells (HCs)/maternal macrophages (MMs), decidual stromal cells (DSCs), and T cells (Figure 2B). Deconvolution estimates (Appendix A) showed that STB was the most abundant trophoblast cell type in VT ETP samples but was significantly decreased in C1-iPL, whereas EVT, CTB, NK and HC/MM cells were more abundant in C1-iPL samples. In decidua, no major differences were observed between controls and C2-iPL, although an increase in MM cells was noted in iPL. The outlier DB ETP 1, sample 3, was excluded from further analyses (Figure 2C).

### 3.3. Gene Set Enrichment Analysis

We performed Gene Set Enrichment Analysis (GSEA) comparing iPL samples with ETP control samples for both villi (Figure 3A–C) and decidua (Figure 3D,E). GSEA used hallmark annotations to assess pathway enrichment between the iPL and control groups (FDR ≤ 0.05, Appendix A). Commonly enriched pathways included ‘TNFa signaling via NFKb’ and ‘Inflammatory response’ when comparing our cases with each control group, regardless of the sample tissue type (villi or decidua). Cell cycle-related pathways such as ‘Mitotic spindle’, ‘P53 Pathway’, ‘Apoptosis’ and ‘G2M checkpoint’ were also enriched. Additionally, the ‘MYC Targets’ term was consistently enriched in villi comparisons. These findings suggest that similar molecular pathways may underlie unexplained pregnancy loss, regardless of tissue type.

### 3.4. Differential Expression Analysis Between Idiopathic Pregnancy Loss and Elective Termination of Pregnancy

Following the observation of similar enriched pathways in our iPL cases, regardless of the control group used, we proceeded with a detailed analysis of DEGs for each comparison. In villi samples, when iPL was compared with VT1 controls, we identified 1599 downregulated and 2114 upregulated genes, whereas in the comparison with VT2, we found 905 downregulated and 1079 upregulated genes, and comparison with VT3 revealed 1779 downregulated and 1751 upregulated genes (FDR < 0.05 and log2FC|>2|) (Appendix A). Analysis of the genes that overlapped between these three comparisons resulted in 309 overlapping downregulated DEGs (Figure 4A) and 444 shared upregulated DEGs (Figure 4B).

GO analysis was then conducted separately for overlapping upregulated and downregulated DEGs using gprofiler2, with significant terms defined by an adjusted *p*-value < 0.05 (Appendix A). Downregulated DEGs were significantly enriched in several GO terms related to reproductive and endocrine processes, as well as growth factor activity. Notably, hormone activity was the most enriched molecular function in the iPL, which was associated with genes such as the Growth Hormone (*GH2*), Chorionic Somatomammotropin Hormones (*CSH1*, *CSH2*, and *CSHL1*), Chorionic Gonadotropin (*CGB3*, *CGB8*, *CGB5*, and *CGB7*), Corticotropin-releasing Hormone (*CRH*), and Glycoprotein Hormones (*CGA*). In addition, the REAC analysis also revealed pathways like ‘Glycoprotein hormones’, ‘Peptide hormone biosynthesis’, and ‘Metabolism of steroid hormones’ as well as terms related to ‘Cell surface interactions at the vascular wall’ and ‘Hemostasis’ (Figure 4C). In contrast, upregulated DEGs in iPL were significantly enriched in pathways such as ‘Antigen processing and presentation of peptide antigen’, ‘Regulation of response to stimulus’, ‘MHC protein complex assembly’, and ‘Positive regulation of cellular process’. These pathways were significantly linked to upregulated genes, such as Major Histocompatibility Complex (MHC) Class I genes (*HLA-A* and *HLA-F*), MHC Class II genes (*HLA-DRA*, *HLA-DOA*, *HLA-DPB1*, *HLA-DMA*, and *HLA-DQB2*), and genes critical to the MHC class I-dependent antigen presentation pathway (*TAP2* and *TAPBP*). The REAC pathways enriched in these upregulated DEGs included ‘Interferon gamma signaling’, ‘Cytokine Signaling in Immune system’, and ‘Interlekin-10 signaling’ (Figure 4D).

Regarding decidua cells, only 26 overlapping downregulated DEGs were identified (Figure 5A). Consequently, we performed GO analysis comparing iPL with DB ETP 1 (Figure 5B) and with DB ETP 2 control groups (Figure 5C), but the biological processes differed between the two comparisons. Indeed, when we compared with DB ETP 1, all the terms were related to the cell cycle, whereas when we compared with DB ETP 2, ‘Oxidative phosphorylation’, ‘transmembrane transport’, and ‘cellular respiration’ were the terms retrieved. This discrepancy may be due to the small size of the control groups, particularly DB ETP 1, which included only two samples. However, we identified 664 shared upregulated DEGs (Figure 5D), and the biological processes were also related to the immune system process and response (Figure 5E).

### 3.5. Histological Findings

Histological sections from different regions (including the embryo, yolk sac, gestational sac/membranes, chorionic villi, intervillous space, endometrium/decidua, and the implantation site) were analyzed to detect morphological changes potentially linked to transcriptome analysis (Appendix A). All the samples evaluated were well preserved, without associated degenerative changes. In C1 samples, the hematoxylin and eosin (H&E) staining revealed the decidua with predominantly lymphohistiocytic inflammatory infiltrate (Figure 6A,C), with macrophages being evidenced by the immunohistochemical study (Figure 6B,D). Additionally, these samples also showed macrophages infiltration into the intervillous space and fibrin depositions, characteristic of chronic histiocytic intervillositis (CHI) (Figure 6E–H). These findings are indicative of chronic inflammatory changes, suggesting possible maternal immunological dysregulation. In contrast, C2 samples exhibited acute villitis and intervillositis in the intervillous space in three samples, and a decidua with a predominantly neutrophilic inflammatory infiltrate, suggestive of infection (Appendix A). Two other cases showed different morphological characteristics, as one exhibited an increase in STB knots and other hypoxia-associated changes, while the other case showed focal stromal hemorrhage and fibrin deposits. Regarding C2 samples, the histological findings were less consistent across the samples. Moreover, the small and randomly selected sections observed might have led to the omission of potential morphological alterations in other regions.

## 4. Discussion

A successful pregnancy requires intricate interactions between the maternal decidua and fetoplacental unit. Although recurrent pregnancy loss (RPL) has been linked to failures in endometrial decidualization, placental dysfunction, and disruptions in the immune microenvironment at the maternal–fetal interface, the precise etiopathophysiological mechanisms of RPL remain elusive [29]. Here, we analyzed gene expression profiles related to idiopathic pregnancy loss (iPL), aggregating transcriptomic analysis with histological findings.

The initial clustering and deconvolution analysis of all the samples revealed that they aggregate into two groups, one enriched in trophoblast cells and the other in decidual cells, with both being localized separately from the control ETP (Elective Termination of Pregnancy) samples in the PCA plot. In trophoblast-enriched samples, deconvolution analysis revealed a significant reduction in STB in iPL cases compared to ETP controls, while CTB and EVT were increased. This shift may reflect either a compensatory trophoblast response to ensure successful implantation, or a tendency for CTBs to differentiate into EVTs rather than fusing to form STBs, potentially impairing placental function in iPL. Additionally, the observed increase in NK and HC/MM cells in the iPL cases, suggests an immune response. Despite NK cells being the predominant cell type in decidua across the groups, no significant differences in cell type proportions were observed. Notably, NK cells constitute the largest immune cell population in human uterine decidua, comprising approximately 70% of maternal resident lymphocytes, whereas they comprise less than 15% of circulating lymphocytes [30].

Comparative GO analyses with published control datasets consistently showed enrichment of molecular pathways, particularly TNFα signaling via NFκB and inflammatory response pathways. Arutyunyan et al. sought to identify the major regulatory phases driving EVT differentiation by extracting transcription factors that are differentially expressed and active along the EVT differentiation trajectory [12]. The early EVTs, formed from the cytotrophoblast cell columns, show upregulation of NFκB pathway genes (*NFKB2* and *BACH2*) and AP-1 signaling genes (*JDP2* and *ATF3*), potentially leading to epithelial–mesenchymal transition. The activation of the NF-κB pathway is maintained throughout EVT differentiation, but there is upregulation of the NF-κB inhibitor (*NFKBIZ*) during the interstitial EVT stage, which invades the decidual stroma [12]. In our analysis, TNFα signaling via NF-κB was enriched across all three comparisons, and epithelial–mesenchymal transitions were also enriched in comparison with ETP VT2 and VT3 controls. We also observed an upregulation of *NFKB2* and *ATF3* when comparing our cases with ETP VT3 controls, and an upregulation of *BACH2* when compared with ETP VT1 and VT2 controls. Additionally, *NFKBIZ* was a commonly upregulated DEG across all comparisons. These findings support the increased presence of EVTs in iPL cases, suggesting that these EVTs are at different stages of the differentiation process.

The ‘MYC Targets’ was another consistently enriched GO term across all trophoblast comparisons. MYC proteins, crucial in transcriptional regulation and cell proliferation during placental development, have been implicated in immune modulation [31]. However, recent data suggest that MYC may also be a crucial player in immune response. The immune tolerance established during embryo implantation is thought to involve the regulation of immune responses through mechanisms that include the expression of MYC targets genes [32]. Enrichment of MYC targets in iPL cases implies that trophoblasts may engage in defensive immune responses, supporting MYC’s role in immune privilege at this interface. Prior work has linked miRNAs with RPL, highlighting TGF-β signaling’s involvement and, driven by the several upregulated miRNAs that targeted the MYC gene, corroborating our results [33].

Furthermore, transcriptome analysis of the first trimester placental chorionic villi showed binding sites for E2F transcription factors in two-thirds of the DEGs [34]. In line with these findings, we also observed enrichment of E2F targets in iPL cases, compared with ETP VT2 controls. E2F plays a key role in maintaining trophoblastic cell function and placental development, with its disruption potentially leading to placental dysfunction and fetal loss [35]. E2F binding sites, abundant in genes associated with cell cycle and DNA replication, has been demonstrated in the regulation of the mammalian endocycle, contributing to polyploid placental syncytiotrophoblast formation [36]. However, E2F targets enrichment was not observed in comparison with other control groups. It is important to note that the study by Sõber and collaborators included only two cases, which limits the conclusions that can be drawn. [34]. Moreover, transcriptome analysis of ETP samples by Sõber et al. identified an abundance of genes involved in placental hormone secretion, although the study did not specifically compare gene expression levels between ETP and RPL [34]. In our analysis of differentially expressed genes (DEGs) between control and iPL groups, we identified 309 commonly downregulated genes in the villi samples, primarily linked to endocrine pathways, including growth hormone regulation, gonadotropin secretion, and glycoprotein hormones. These pathways are closely linked to the STB function, as these hormones are produced exclusively by STB. This aligns with our previous findings of reduced STB marker expression in iPL cases in third-trimester samples compared to term or known-cause placentas [21], suggesting that impaired syncytium formation may contribute to pregnancy loss irrespective of gestational age. Our study also raises questions about the relationship between STB dysfunction and maternal immune cells, possibly reflecting a process of fetal rejection or abnormal syncytialisation. STB dysfunction may either trigger or result from maternal immune attack, compromising placental function and pregnancy. Syncytium formation and maintenance, tightly regulated and complex, may also enable early recognition of danger signals and activation of innate immune responses [37]. In one of the control studies (ETP VT2 samples), ETP samples compared with RPL, showed that lnc-SLA4A1-1, an enhancer RNA (eRNA), was upregulated in unexplained RPL, where it recruits NF-κB and binds to *CXCL8* promoter, upregulating *CXCL8*. This *CXCL8* increase leads to TNF-α and IL1β release, potentially triggering an immune response and contributing to the unexplained RPL pathogenesis [25]. Our data also supported these findings, showing TNFa signaling via NF-κB enrichment and *CXCL8* upregulation in iPL cases across all control groups, including villi and decidua samples. Chemokines like *CXCL8* play essential roles in intercellular communication and signal transduction at the maternal–fetal interface, participating in embryo implantation and trophoblast invasion [38] and promoting cell migration and invasion, consistent with its role in the normal EVT function [39]. Another study analyzing genome-wide DNA methylation and DEGs from placental villi from RPL patients versus controls revealed that hypomethylation near the *PRDM1* transcription start site upregulates its expression, promoting apoptosis and trophoblast migration in RPL. Additionally, DEGs were enriched in immune response pathways, highlighting immune dysregulation’s role in RPL [40]. Similarly, we observed *PRDM1* upregulation in iPL cases compared with ETP VT1 controls, emphasizing immune involvement.

Among the GO terms related to commonly upregulated genes across the three comparisons, we identified 444 DEGs associated with immune response, including MHC protein complex, antigen presentation and processing, and the regulation of response to stimuli. Huang et al. observed similar findings in RPL, noting antigen processing and presentation as an enriched pathway of DEGs, alongside biological processes linked to immune system function such as leukocyte adhesion and proliferation, cellular extravasation, eosinophil migration, and MHC class II protein complex involvement [41]. The upregulation of these immune-related pathways in iPL aligns with the increased EVT proportion observed, as EVTs differentiate from cytotrophoblast progenitors, invade the uterine wall, and interact with maternal immune cells to modulate immune tolerance at the maternal–fetal interface [42]. The maternal immune system balances the sufficient reactivity for pathogen defense with tolerating paternal alloantigens to maintain fetal integrity. Unlike most cells, EVTs do not express ubiquitous classical MHC class I molecules (i.e., HLA-A and HLA-B) and MHC class II molecules (HLA-DP, HLA-DQ, and HLA-DR), which are typically the targets by alloreactive T cells in transplant rejection. Instead, they express HLA-C and the nonclassical MHC class I molecules, such as HLA-E, HLA-G, and possibly HLA-F [43,44,45,46,47]. HLA-G on EVTs binds to decidual NK cell receptors, activating the NF-κB pathway, which leads to the production of cytokines, promoting vascular permeability, angiogenesis, and EVT [48]. Additionally, HLA-G has been shown to enhance fetal growth by inducing stimulating NK cell secretion of growth-promoting factors [49,50]. Following the observed increase in EVTs, we also observed an overexpression of HLA-C in iPL cases compared to the ETP VT1 and VT2 control groups, alongside upregulation of non-classical isoforms, HLA-E, HLA-F, and HLA-G. However, we also observed atypical overexpression of HLA-A in iPL cases as well as HLA-B (comparing to ETP VT1 and VT3 control groups) and MHC II isoforms, including HLA-DP, HLA-DQ, and HLA-DR. The prevailing hypothesis posits that aberrant HLA-A or HLA-B expression in trophoblasts activates CD8+ T cells, potentially contributing to fetal rejection [51]. Since CTBs are highly proliferative and act as the “stem cells” of the placenta, which can originate STBs and EVTs, it appears that in iPL cases, a deviation in the CTB differentiation pathway occurs, resulting in impaired STB formation and increased EVT formation.

In contrast, only 26 shared downregulated DEGs were identified in decidual samples between controls and iPL, possibly due to limited sample size; however, 664 upregulated DEGs were shared, primarily associated with maternal immune response. Wang et al. used multi-omics to characterize villi and decidua in RPL. The DEGs identified in villi were primarily associated with cell growth and development, although these were based on only two datasets of microarrays. Conversely, the DEGs in decidua retrieved from Yu et al.’s dataset (one of our control groups, DB ETP 2) were related to nutrient transport, immune response, extracellular matrix remodeling, and angiogenesis, indicating active immune responses in decidua due to maternal–fetal interactions [52]. Supporting the role of immune dysfunction, Chen et al. profiled decidual immune cells from normal and idiopathic RPL pregnancies using single cell RNA-Seq, showing immune variations in RPL pathogenesis [53]. Our study stands out by integrating transcriptomic and histological data from iPL cases, providing deeper insights into the placental mechanisms and pathogenesis of iPL, complementing recent transcriptomic studies of trophoblasts and decidua [26,29,54,55].

Histological analysis of decidual samples showed heterogeneity, identifying three distinct histological patterns. The most common pattern (three of five samples) displayed acute inflammatory changes, suggesting recent infection, supporting GO analysis findings of immune system regulation upregulation. Histological findings also align with our hypothesis of a maternal immune response in villi samples from idiopathic cases, as we observed a predominantly lymphohistiocytic inflammatory infiltrate and fibrin deposition, characteristic of chronic histiocytic intervillositis (CHI) [56]. Although a few studies have examined placental protein expression in CHI, upregulation of intercellular adhesion molecule-1 (ICAM-1) on STBs has been reported [57]. ICAM-1 mediates leucocytes migration and enhances monocyte adhesion to the syncytial surface. In CHI, elevated ICAM-1 expression may be either contributing to maternal immune cell recruitment into the intervillous space or be a consequence of placental damage. Despite reduced STB, *ICAM-1* remained upregulated in iPL cases, suggesting CHI presence, which has been associated with miscarriage and stillbirth risk [58,59]. Another factor implicated in CHI is CD200, an immunosuppressive protein promoting M2 macrophages and Treg differentiation while inhibiting cytotoxic NK cells [60,61]. Reduced CD200 expression has been observed in early pregnancy losses and CHI villi, suggesting a mechanism for impaired immune tolerance [60]. We also found a significant reduction in *CD200* expression in iPL. CHI’s recurrence risk in future pregnancies ranges from 25 to 100%, and its diagnosis can only be confirmed post-delivery through placental examination as no biomarkers are currently available. [62]. The mechanism by which CHI contributes to adverse outcomes remains unclear but is hypothesized to involve an inappropriate immune response to the semi-allogeneic fetus, consistent with our findings.

The study’s main limitations were the size sample and the lack of a control group collected concurrently with cases. To address this, we incorporated independent control groups to strengthen the robustness of the comparative analysis, although we recognize that variability between the controls cannot be entirely eliminated. In future studies, increasing the number of samples across groups will be essential to confirm these findings. Despite these limitations, our findings suggest potential molecular pathways that may be relevant for couples experiencing RPL. Immune modulation dysfunction has been proposed as a cause of miscarriage [63]; however, the physiological mechanisms that enable maternal tolerance of paternal antigens remain poorly understood, and the molecular mechanisms underlying RPL due to immune dysregulation are still elusive. Profiling gene expression and biological pathways involved in pregnancy failures could help identify novel biomarkers or therapeutic targets relevant to RPL management. Nonetheless, the complexity of pregnancy loss presents a challenge in determining whether the altered pathways observed are causative mechanisms or secondary to the miscarriage process.

## 5. Conclusions

Our study provides new insights into immune dysregulation in iPL, particularly through the altered balance between STB and EVT differentiation and its effect on maternal immune responses. By integrating transcriptomic and histological data, we offer a comprehensive, tissue-level perspective that goes beyond previous RNA sequencing studies. This multilayered approach allows us to link molecular alterations to histological changes, providing a more complete understanding of the mechanisms underlying iPL. Importantly, our findings suggest that disruptions in trophoblast differentiation and immune regulation may play a central role in pregnancy loss. Further research is needed to confirm these findings, elucidate the underlying molecular mechanisms, and explore potential diagnostic or therapeutic strategies that could improve clinical management of RPL.

## Figures and Tables

**Figure 1 biology-14-00811-f001:**
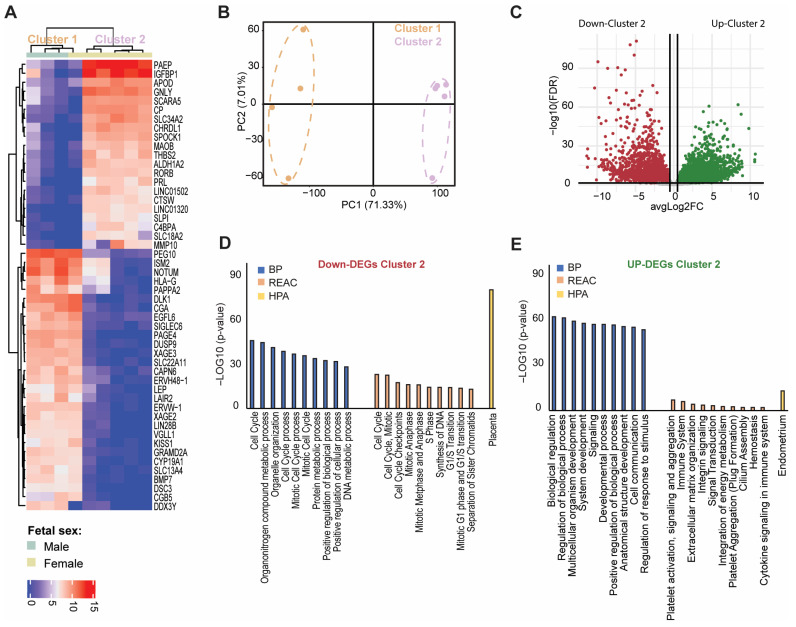
Hierarchical clustering and gene expression analysis of idiopathic pregnancy losses (iPL). Heatmap of the top 50 most highly variable genes across nine products of conception (POC) (**A**) and PCA plot for clustering gene expression data (**B**). Red represents genes with high expression levels, while blue represents genes with low expression levels. Volcano plot of gene expression analysis between C1 and C2 (**C**). The thick horizontal line represents the threshold of FDR < 0.05. Colour coding is based on this threshold and represents the differentially expressed genes (DEGs) between the respective clusters. Red dots indicate significantly down-expressed genes, while green dots indicate significantly up-expressed genes. Gene Ontology (GO) enrichment analysis of biological processes (BP), reactome pathway (REAC) and human protein atlas (HPA) were presented in the bar graph, regarding down-regulated genes (**D**) and up-regulated genes (**E**). Top 10 of GO for BP and REAC categories sorted by decreasing order of *p*-values, based on the GO enrichment test *p*-value.

**Figure 2 biology-14-00811-f002:**
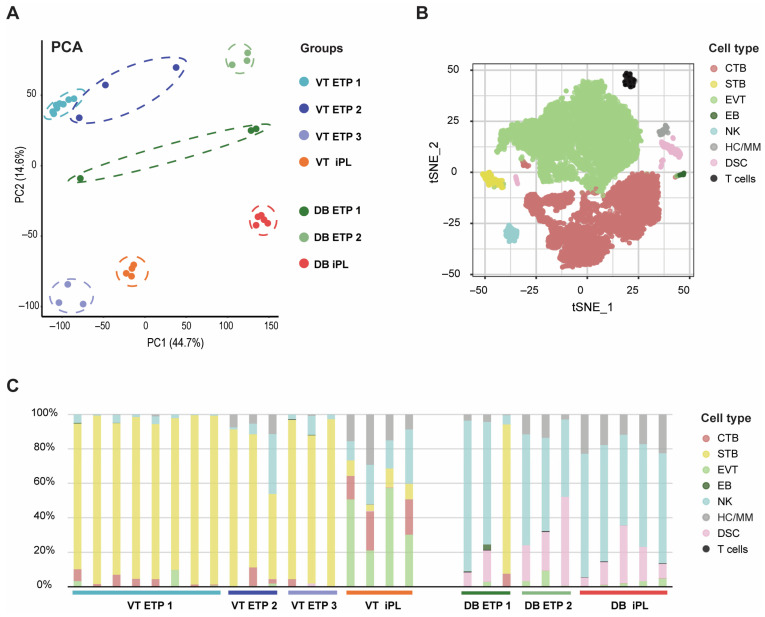
Principal Component Analysis, TSNE Annotation, and Bulk RNAseq Deconvolution of samples from idiopathic pregnancy loss (iPL) and Elective Termination of Pregnancy (ETP). Principal component analysis (PCA) plot of the expression profile of all samples revealed a significant separation between iPL and ETP samples (**A**). TSNE annotation was performed to delineate the distinct cell types within these clusters, namely, cytotrophoblasts (CTB), syncytiotrophoblasts (STB), extravillous trophoblasts (EVT), erythroblasts (EB), natural killer (NK) cells, Hofbauer cells (HC)/ maternal macrophages (MM), decidual stromal cells (DSC) and T cells (**B**). The deconvolution estimates of the bulk RNAseq data are presented in percentage. Each bar represents a sample, which are grouped by the horizontal bars corresponding to each biological group (**C**).

**Figure 3 biology-14-00811-f003:**
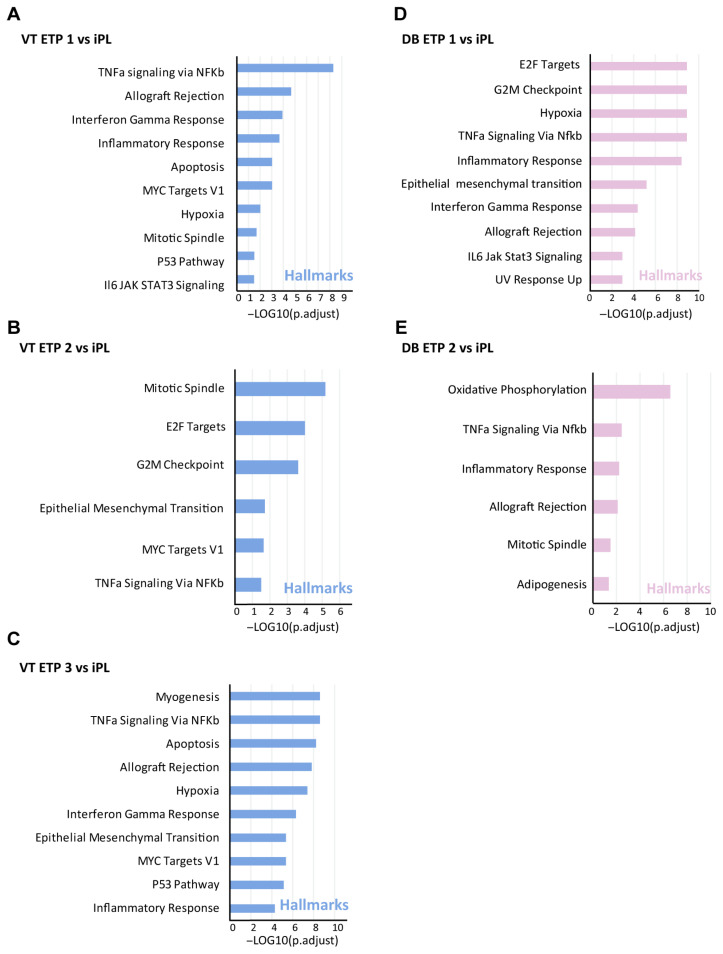
GSEA enrichment of RNAseq data identifies the related signaling pathways in idiopathic pregnancy loss (iPL). Results of GSEA Hallmark analysis showing enriched gene sets for different comparisons of villi (**A**–**C**) and decidua (**D**,**E**). Bars depict the enrichment score for the top 10 most enriched pathway sets, the blue bars indicate significant enrichment hallmarks at FDR < 0.05 of villi companions, while pink bars regarding decidua comparisons.

**Figure 4 biology-14-00811-f004:**
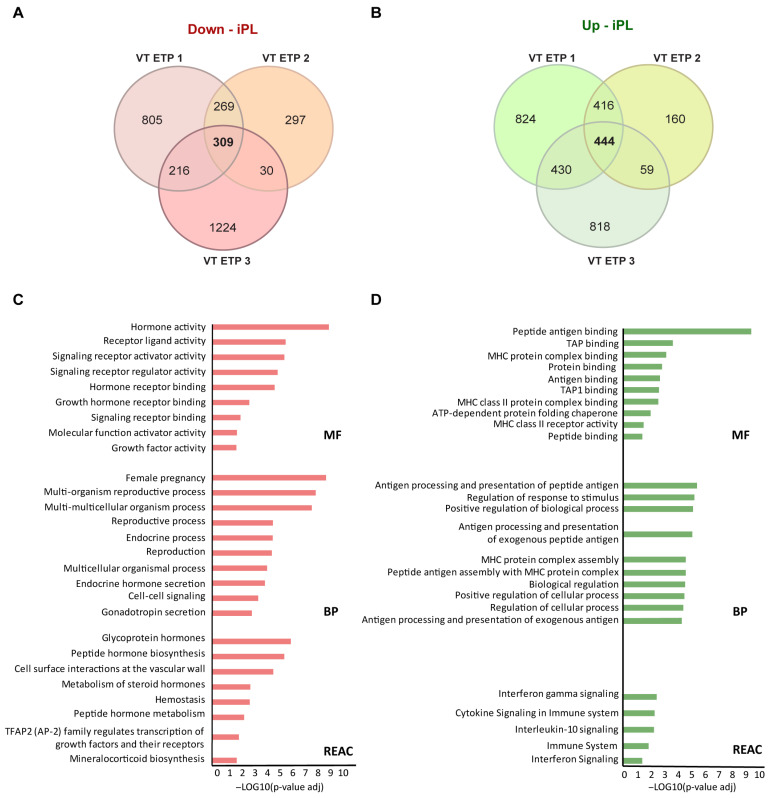
Analysis of Differentially Expressed Genes (DEGs) and Gene Ontology (GO) analysis of villi samples. Venn diagrams showing the overlap of DEGs among the three comparisons studied: iPL vs. VT ETP 1, iPL vs. VT ETP 2 and iPL vs. VT ETP 3. The red Venn diagram corresponds to downregulated DEGs (**A**), while the green Venn diagram corresponds to upregulated DEGs (**B**). GO analysis results showed that downregulated DEGs (red bars, **C**) and upregulated DEGs (green bars, **D**) were categorized in molecular functions (MF), biological processes (BP) and Reactome pathway (REAC). The X-axis indicates the –Log10(*p*-value adj), and the Y-axis indicates the GO terms.

**Figure 5 biology-14-00811-f005:**
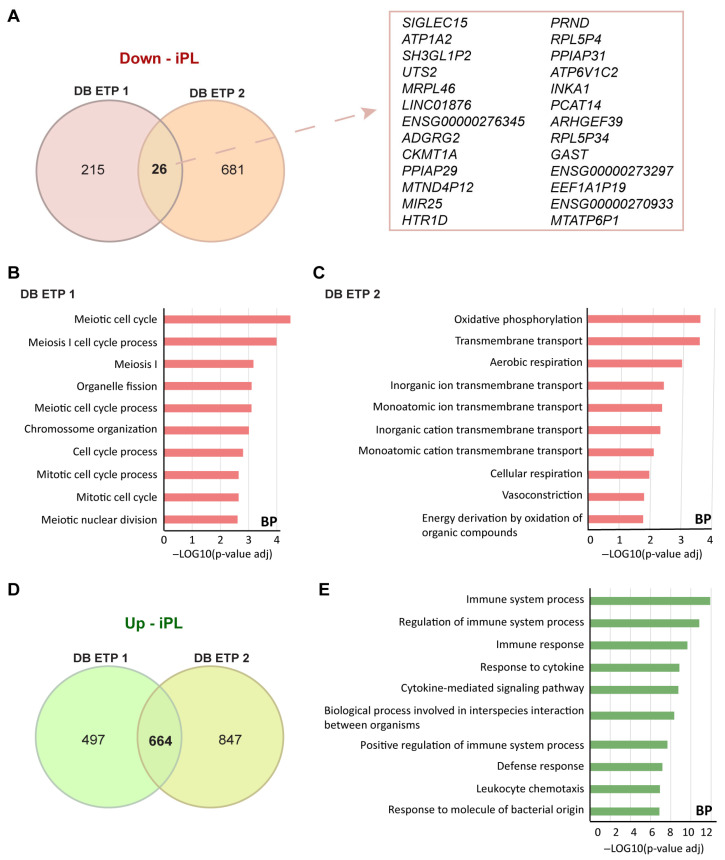
Analysis of Differentially Expressed Genes (DEGs) and Gene Ontology (GO) analysis of decidua samples. Venn diagrams showing the overlap of DEGs among the two comparisons studied: iPL vs. DB ETP 1 and iPL vs. DB ETP 2. The red Venn diagram corresponds to downregulated DEGs and the shared genes are shown in the right window (**A**). GO analysis results from iPL vs. DB1 (**B**) and iPL vs. DB ETP 2 (**C**) were categorized in molecular functions (MF), biological processes (BP) and Reactome pathway (REAC). The green Venn diagram corresponds to upregulated DEGs (**D**) and the GO analysis results showed that upregulated DEGs (**E**) were categorized in molecular functions (MF), biological processes (BP) and Reactome pathway (REAC). The X-axis indicates the –Log10(*p*-value adj), and the Y-axis indicates the GO terms.

**Figure 6 biology-14-00811-f006:**
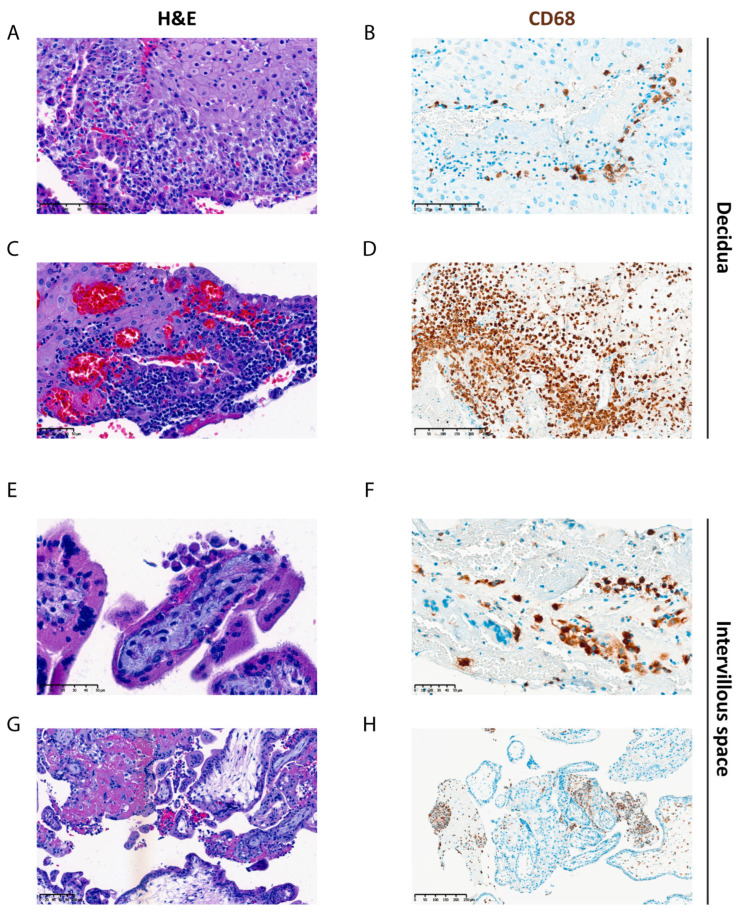
Histological (H&E) and immunohistochemical (CD68) staining of placentas from Idiopathic pregnancy losses (iPL). Decidua showing lymphohistiocytic inflammatory infiltrate (**A**,**C**) and the macrophages being evidenced by the immunohistochemical study (**B**,**D**). Intervillous space showing aggregation of macrophages and fibrin accumulation shown by H&E staining (**E**,**G**), with macrophages being highlighted by immunohistochemistry (**F**,**H**).

**Table 1 biology-14-00811-t001:** Sample characteristics. iPL—idiopathic pregnancy loss; F—female; M—male; G—gravida; P—para (parity); A—abortion.

Sample ID	Maternal Age (Years)	Fetal Sex	Gestational Age (Weeks)	Previous Obstetric History
**iPL 1**	37	F	9	1G
**iPL 2**	25	F	10	2G 1A
**iPL 3**	16	M	9	2G 1A
**iPL 4**	37	F	9	4G 3A
**iPL 5**	30	M	9	8G 7A
**iPL 6**	37	M	6	11G 10A
**iPL 7**	32	F	8	4G 1P 2A
**iPL 8**	34	F	8	2G 1A
**iPL 9**	40	F	9	4G 3A

## Data Availability

Data is provided within the manuscript and in Appendix A. Additional data can be provided on request.

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
