# Peer review of "Immune Dysregulation and Trophoblastic Dysfunction as a Potential Cause of Idiopathic Recurrent Pregnancy Loss"

_biology, 2025, doi:10.3390/biology14070811_

Round 1
Reviewer 1 Report
Comments and Suggestions for Authors
The present study examined the transcriptome of human products of 129 conception from idiopathic pregnancy loss (iPL) in early stages of pregnancy. The study is interesting and provides useful information. However, there are some comments should be addressed.
Abstract
L56: the sample size seems to be low, did the authors perform a power analysis to determine the suitable sample size for conclusive outcomes.
L69-70: “Our results provide insights into iPL pathogenesis, suggesting potential biomarkers and therapeutic strategies to address RPL”. the study doesn’t examine therapeutic strategies, please mentioned only what the study examined.
L71: mention only pregnancy loss instead of idiopathic and recurrent. That will be enough for appearing the manuscript in the search engines especially these words are mentioned in the title.
Introduction
- Please define clearly the research gab and hypothesis
Methods.
L152: correct the wors “pout”.
L156-157: The sentence is confusing, please clarify.
- Please provide even a single histopathological image for more clarification.
- Please add a separate section for the statistical analysis
Results
- There are illustrative tables or figures.
- The section is not well-presented or -organized
Discussion
- This section is too long and confounding, please be concise.
- Please add a separate section to the conclusion after the end of the discussion section.
Author Response
We are truly grateful for the opportunity to resubmit our manuscript entitled " Immune dysregulation and trophoblastic dysfunction as potential cause of idiopathic recurrent pregnancy loss", to Biology of MDPI. We sincerely thank the reviewers for their thoughtful and constructive feedback, which has been crucial in helping us improve the quality of our work. We have carefully considered each comment and concern, and we believe that our revised manuscript addresses the points raised in a comprehensive and rigorous manner. Below, we provide detailed responses to all reviewer comments, aiming to clarify any ambiguities and to highlight the improvements made throughout the manuscript.
Reviewer 1:
The present study examined the transcriptome of human products of 129 conception from idiopathic pregnancy loss (iPL) in early stages of pregnancy. The study is interesting and provides useful information. However, there are some comments should be addressed.
Abstract
L56: the sample size seems to be low, did the authors perform a power analysis to determine the suitable sample size for conclusive outcomes.
We appreciate the reviewer’s comment regarding the sample size. Due to the strict inclusion and exclusion criteria applied in our study — particularly the focus on truly idiopathic cases of recurrent pregnancy loss — the recruitment of a larger cohort with such specific characteristics proved challenging. Nevertheless, the findings obtained were statistically significant for key parameters and are in line with previously reported evidence in the literature. Moreover, we enhanced the robustness of our analysis by conducting in-depth assessments of each individual sample, including deconvolution of transcriptomic data and detailed individual histochemical analyses, which added valuable granularity to the results. While we acknowledge that the limited sample size may impact the generalizability of our findings, we believe that our study offers important insights into a complex and underexplored condition. It also lays the groundwork for future larger-scale investigations. We have now explicitly addressed this limitation in the revised Discussion section (lines 539-544).
L69-70: “Our results provide insights into iPL pathogenesis, suggesting potential biomarkers and therapeutic strategies to address RPL”. the study doesn’t examine therapeutic strategies, please mentioned only what the study examined.
We thank the reviewer for pointing this out. We agree that the original sentence may have overstated the implications of our findings. The study did not directly investigate therapeutic strategies. We have revised the sentence:
“Our results provide insights into iPL pathogenesis, highlighting potential biomarkers that may contribute to improved diagnosis and future research.”
L71: mention only pregnancy loss instead of idiopathic and recurrent. That will be enough for appearing the manuscript in the search engines especially these words are mentioned in the title.
We thank the reviewer for this helpful suggestion. We agree that using the broader term "pregnancy loss" is sufficient here. We have revised the keywords in the manuscript.
Introduction
- Please define clearly the research gab and hypothesis
We thank the reviewer for this valuable suggestion. We fully agree that clearly highlighting the research gap at the end of the Introduction strengthens the rationale of the study and provides a more focused context for the reader. We have revised the Introduction and added the following paragraph.
“Although multiple factors have been implicated in pregnancy loss, the precise cellular and molecular mechanisms underlying idiopathic pregnancy loss (iPL) remain largely un-known. Most existing studies describe transcriptomic alterations in iPL samples but do not explore what these changes imply in terms of cellular identity, trophoblast differentiation, or functional roles.”
Methods.
L152: correct the wors “pout”.
We thank the reviewer for noticing this typographical error. The word “pout” has been corrected to “out” in the revised manuscript.
L156-157: The sentence is confusing, please clarify.
We thank the reviewer for this comment. We agree that the original sentence lacked clarity. Our intention was to explain that, in order to minimize variability and strengthen our comparative analysis, we incorporated control samples from more than one previously published dataset, allowing us to include multiple independent control groups. We have revised the sentence for clarity as follows:
“To address this limitation, we included control samples from more than one previously published dataset, allowing us to include multiple independent control groups.”
- Please provide even a single histopathological image for more clarification.
We sincerely apologize for the omission. We are unsure how this issue occurred during the previous submission, as all files were initially prepared and included both within the manuscript and as separate files upload. Nevertheless, all figures and supplementary materials have now been carefully reviewed and correctly resubmitted with the revised version of the manuscript. The histopathological images can be found in Figure 6.
- Please add a separate section for the statistical analysis
We thank the reviewer for this thoughtful suggestion. After careful consideration, we believe that presenting the statistical methods alongside each respective bioinformatic analysis enhances clarity and helps the reader better follow the sequence, context, and specific criteria applied at each step of the workflow. Given the complexity and layered nature of the analyses, integrating the statistical approaches within each subsection allows for a more logical and accessible presentation of the methods used.
We are concerned that consolidating all statistical information into a single, separate section might reduce transparency and make it more difficult to associate specific analyses with their corresponding results. However, if the reviewer still considers the inclusion of a dedicated Statistical Analysis section essential, we are fully willing to restructure the manuscript accordingly.
Results
- There are illustrative tables or figures.
- The section is not well-presented or -organized
We thank the reviewer for the observation. We believe that the lack of access to the figures and tables in the initial submission may have contributed to the impression that the Results section was not well-presented or organized. In the revised version, all figures and tables have been carefully checked and correctly submitted. We expect that, with the full set of visual results now available, the structure and clarity of the Results section will be significantly improved and more easily understood.
Discussion
- This section is too long and confounding, please be concise.
We thank the reviewer for this comment and understand the concern regarding the length of the Discussion section. However, we respectfully believe that a substantial reduction would compromise the depth and clarity of the manuscript. Given the number of results presented — including six main figures and additional supplementary data — we found it necessary to provide appropriate interpretation and contextualization for each major finding. Eliminate parts of this discussion would risk overlooking relevant biological insights and limiting the scientific value of the work, particularly considering the exploratory nature of the study and its potential to inform future research directions involving larger cohorts and functional validation.
We agree that the addition of a separate Conclusion section was an excellent suggestion, and it has also contributed to a more concise overall Discussion. We remain open to further adjustments if the reviewer has specific suggestions for streamlining any part of the text.
- Please add a separate section to the conclusion after the end of the discussion section.
We thank the reviewer for this helpful suggestion. A separate Conclusion section has been added following the end of the Discussion. To ensure coherence and avoid redundancy, we adapted the final paragraph of the Discussion into this new section, which also allowed us to shorten the Discussion, as previously suggested.
“5. Conclusion
Our study provides new insights into immune dysregulation in iPL, particularly through the altered balance between STB and EVT differentiation and its effect on mater-nal immune responses. By integrating transcriptomic and histological data, we offer a comprehensive, tissue-level perspective that goes beyond previous RNA sequencing stud-ies. This multilayered approach allows us to link molecular alterations to histological changes, providing a more complete understanding of the mechanisms underlying iPL. Importantly, our findings suggest that disruptions in trophoblast differentiation and im-mune regulation may play a central role in pregnancy loss. Further research is needed to confirm these findings, elucidate the underlying molecular mechanisms, and explore po-tential diagnostic or therapeutic strategies that could improve clinical management of RPL.”
Reviewer 2 Report
Comments and Suggestions for Authors
This manuscript investigates the molecular mechanisms underlying idiopathic recurrent pregnancy loss through integrative transcriptomics and histological analyses, proposing immune dysregulation and trophoblastic dysfunction as potential causes. However, some important questions in this manuscript should be addressed.
- All mentioned figures were no found in the manuscript and supplementary files.
- The limited sample size (n=9 iPL cases, 8/9 with RPL) risks bias. Clearly state limitations in the discussion and propose future expansion.
- Reliance on publicly available datasets introduces technical variability (e.g., sequencing platforms, sample processing). Despite the standardization process, the heterogeneity risks between the control group and the experimental group still need to be emphasized.
- CHI diagnosis lacks standardized criteria. limited sample size may affect the universality of the conclusion.
- Although the association between immune dysregulation and trophoblast differentiation has been proposed, the causal relationship between the two remains unclear, and a hypothesis model needs to be constructed.
- The potential biomarkers discovered (such as CXCL8, HLA-G) lack functional verification or clinical correlation analysis (such as thresholds for predicting the risk of miscarriage), recommending to supplement.
- Some abbreviations, such as CHI, were not defined when they first appeared and need to be uniformly corrected.
Author Response
We are truly grateful for the opportunity to resubmit our manuscript entitled " Immune dysregulation and trophoblastic dysfunction as potential cause of idiopathic recurrent pregnancy loss", to Biology of MDPI. We sincerely thank the reviewers for their thoughtful and constructive feedback, which has been crucial in helping us improve the quality of our work. We have carefully considered each comment and concern, and we believe that our revised manuscript addresses the points raised in a comprehensive and rigorous manner. Below, we provide detailed responses to all reviewer comments, aiming to clarify any ambiguities and to highlight the improvements made throughout the manuscript.
Reviewer 2:
This manuscript investigates the molecular mechanisms underlying idiopathic recurrent pregnancy loss through integrative transcriptomics and histological analyses, proposing immune dysregulation and trophoblastic dysfunction as potential causes. However, some important questions in this manuscript should be addressed.
1. All mentioned figures were no found in the manuscript and supplementary files.
We sincerely apologize for the omission. We are unsure how this issue occurred during the previous submission, as all files were initially prepared and included both within the manuscript and as separate files upload. Nevertheless, all figures and supplementary materials have now been carefully reviewed and correctly resubmitted with the revised version of the manuscript.
2. The limited sample size (n=9 iPL cases, 8/9 with RPL) risks bias. Clearly state limitations in the discussion and propose future expansion.
We appreciate the reviewer’s comment regarding the sample size. Due to the strict inclusion and exclusion criteria applied in our study, particularly the focus on truly idiopathic cases of recurrent pregnancy loss, recruiting a larger cohort with such specific characteristics was particularly challenging. We have now explicitly addressed this limitation in the revised Discussion section and proposed directions for future research (lines 539-544).
“The study's main limitations were the size sample and the lack of a control group collected concurrently with cases. To address this, we incorporated independent control groups to strengthen the robustness of the comparative analysis, although we recognize that heterogeneity between the controls cannot be entirely eliminated. In future studies, increasing the number of samples across groups will be essential to confirm these findings. Despite these limitations, our findings suggest potential molecular pathways that may be relevant for couples experiencing RPL. “
3. Reliance on publicly available datasets introduces technical variability (e.g., sequencing platforms, sample processing). Despite the standardization process, the heterogeneity risks between the control group and the experimental group still need to be emphasized.
We thank the reviewer for raising this important point. We fully acknowledge that incorporating control samples from publicly available datasets may introduce technical variability, including differences in sequencing platforms and sample processing protocols. Although we applied rigorous normalization and data integration procedures to minimize such variability, we recognize that heterogeneity between the controls cannot be entirely eliminated. To mitigate this risk, we included three independent control groups rather than relying on a single dataset, which could introduce dataset-specific biases. The use of multiple control groups—each consisting of samples from elective terminations of pregnancy—allowed us to compare our findings across independent datasets and verify the consistency of differential expression patterns. Furthermore, we applied the same bioinformatic processing pipeline to both publicly available raw data and our own samples, ensuring uniformity in quality control, normalization, and analysis. The transcriptomic profiles of the control samples were also found to be relatively homogeneous, which further increased our confidence in the robustness of the comparative results. We have also further emphasized this limitation in the Discussion section (lines 539-544).
4. CHI diagnosis lacks standardized criteria. limited sample size may affect the universality of the conclusion.
We thank the reviewer for this important observation. The diagnosis of chronic histiocytic intervillositis (CHI) in this study was performed by a specialist in anatomical pathology, following established international histopathological criteria and in accordance with the institutional protocols of our hospital. To ensure diagnostic consistency, all samples were independently reviewed and the diagnosis was confirmed by Dr. Ana Braga, a board-certified anatomical pathology specialist and co-author of this manuscript. By applying standardized internal procedures and confirming diagnoses through expert review, we aimed to ensure the highest possible diagnostic reliability within our cohort. Regarding the limited sample size, we fully recognize that this constrains the generalizability of our findings. Our primary intention was not to draw universal conclusions, but rather to perform a detailed and individual histopathological characterization of a carefully selected and clinically well-characterized samples. Nevertheless, it is noteworthy that, despite the small sample size, the cases displayed overlapping features, suggesting potential shared pathological mechanisms. Even so, this limitation about sample size has been explicitly addressed in the revised Discussion section.
5. Although the association between immune dysregulation and trophoblast differentiation has been proposed, the causal relationship between the two remains unclear, and a hypothesis model needs to be constructed.
We thank the reviewer for this insightful comment. We agree that the causal relationship between immune dysregulation and impaired trophoblast differentiation requires further studies to elucidate the underlying mechanisms of pregnancy loss. We consider that immune dysregulation and trophoblast differentiation may represent independent mechanisms or may even have an additive effect. Additional evidence is needed to clarify which is the cause and which is the consequence. Furthermore, the graphical abstract, included in the submission, illustrates the hypothetical model proposed by our study, highlighting the potential interactions between immune pathways and trophoblast dysfunction.
6. The potential biomarkers discovered (such as CXCL8, HLA-G) lack functional verification or clinical correlation analysis (such as thresholds for predicting the risk of miscarriage), recommending to supplement.
We thank the reviewer for this important observation. We fully acknowledge that the candidate genes identified in our study have not been functionally validated for predicting miscarriage risk. Unfortunately, due to the lack of biological control samples, we were unable to perform additional confirmatory techniques, such as real time qPCR or Western blotting, to validate the differential expression of these potential biomarkers between the cases and controls. However, we do not claim these genes as confirmed biomarkers, but rather as potential candidates worthy of further investigation. Our results provide a crucial exploratory basis for future studies, including functional assays, protein-level analyses, and clinical validation to confirm the biological and diagnostic relevance of the findings.
7. Some abbreviations, such as CHI, were not defined when they first appeared and need to be uniformly corrected.
We thank the reviewer for this note. The manuscript has been thoroughly reviewed.
Round 2
Reviewer 1 Report
Comments and Suggestions for Authors
I would like to thank the authors for addressing the comments in a professional manner.
Reviewer 2 Report
Comments and Suggestions for Authors
The author has revised the article in accordance with the requirements. I have no further questions ,but all figures are still not seen in this manuscript and supplementary file.